# Combined Transcriptomics and Metabolomics Analysis Reveals the Effect of Selenium Fertilization on *Lycium barbarum* Fruit

**DOI:** 10.3390/molecules28248088

**Published:** 2023-12-14

**Authors:** Wenqin Yang, Tingting Jiang, Yaqi Wang, Xiaojing Wang, Rui Wang

**Affiliations:** 1College of Agronomy, Ningxia University, Yinchuan 750021, China; 12022131566@stu.nxu.edu.cn (W.Y.); jiangtingting0321@126.com (T.J.); wangyaqi517@foxmail.com (Y.W.); 2Ningxia Research Institute of Quality Standards and Testing Technology of Agricultural Products, Yinchuan 750001, China

**Keywords:** exogenous selenium, *Lycium barbarum*, differential metabolites, differential genes

## Abstract

As a beneficial nutrient and essential trace element, selenium plays a significant role in plant growth functions and human protein biosynthesis. Plant selenium enrichment is mainly obtained from both natural soil and exogenous selenium supplementation, while human beings consume selenium-enriched foods for the purposes of selenium supplementation. In this study, different types of selenium fertilizers were sprayed onto *Lycium barbarum* in Ningxia, and transcriptomics and metabolomics techniques were used to explore the effects of selenium on the fruit differentials and differential genes in *Lycium barbarum*. Taking the “Ning Qiyi No.1” wolfberry as the research object, sodium selenite, nano-selenium, and organic selenium were sprayed at a concentration of 100 mg·L^−1^ three times from the first fruiting period to the harvesting period, with a control treatment comprising the spraying of clear water. We determined the major metabolites and differential genes of the amino acids and derivatives, flavonoids, and alkaloids in ripe wolfberries. We found that spraying selenium significantly enhanced the *Lycium barbarum* metabolic differentiators; the most effective spray was the organic selenium, with 129 major metabolic differentiators and 10 common metabolic pathways screened after spraying. Nano-selenium was the next best fertilizer we screened, with 111 major metabolic differentiators, the same number as organic selenium in terms of differential genes and common metabolite pathways. Sodium selenite was the least effective of the three, with only 59 of its major metabolic differentials screened, but its differential genes and metabolites were enriched for five common pathways.

## 1. Introduction

*Lycium barbarum* has been found to be rich in various nutrients that are required by the human body, such as polysaccharides, amino acids, vitamins, etc., while also being a rich source of antioxidant compounds [1,2]. Selenium plays an important role in a variety of metabolic processes; in appropriate amounts, selenium promotes the growth of animals and plants, and excessive selenium not only affects the growth of animals and plants, but also causes toxic symptoms [3,4,5]. Selenoproteins involved in metabolism have the same redox potential as potassium proteins and are also the most important pathway through which selenium can exert its effects. Selenium significantly promotes plant growth, enhances plant resistance, improves plant quality, and acts as a yield enhancer [6,7]. Selenium comes from a wide range of sources, mainly in the chemical form of sodium selenite, but also through the yeast assimilation of yeast organic selenium and via synthesis through nanotechnology nano-selenium, etc. Plant access to selenium is mainly achieved through the root system, being absorbed from the soil into the body; the same plants’ leaves also have the ability to directly and rapidly absorb nutrients and transport them to other tissues and organs. This method does not directly come into contact with the soil, reducing soil fixation; it is, thus, more conducive to improving the utilization of nutrients [8,9]. Selenium is absorbed by the leaves of *Lycium barbarum* and is assimilated and absorbed into the plant via selenate reduction, selenoamino acid synthesis, and metabolic pathways for volatile selenium formation [10,11].

According to selenium biotransformation studies conducted by G. Lucidum, inorganic selenium can be biotransformed into water-soluble selenium, which exhibits three times more superoxide and hydroxyl radical scavenging than the original protein, and its antioxidant effect is significantly and positively correlated with selenium content [12]. Through the base application and foliar spraying of selenium fertilizer on wolfberries, White et al. concluded that the foliar spraying of selenium fertilizer was more effective than the base application of selenium fertilizer, enhancing the conversion rate of organic selenium in the dried fruits of wolfberry plants [13]. Spraying an appropriate amount of selenium fertilizer can significantly increase the accumulation of selenium and other trace elements in *Lycium barbarum*; when spraying selenium fertilizer in the appropriate concentration range, the more selenium applied, the stronger the selenium enrichment abilities in *Lycium barbarum*. This promotes the growth of *Lycium barbarum*, enhancing its quality and increasing its yield [14,15]. By analyzing different forms of selenium in selenium-enriched wolfberries, Ma et al. showed that organic selenium existed in the form of proteins in most of the fruits, and only a small portion existed in the form of RNA and polysaccharides [16]. Shinde et al. analyzed and identified genes related to the chlorophyll content and relative water content of pearl millet as positive regulators of RWC (relative water content) by conducting a transcriptomic analysis [6]. In the study conducted by Shi et al., a metabolomic analysis of the response to nitrogen levels in *Lycium barbarum* was carried out using the UPLC-ESI-MS/MS method; in total, 612 metabolites, including amino acids, flavonoids, carbohydrates, organic acids, and lipids, were identified in the metabolomic analysis at the three nitrogen levels, of which 53 metabolites (lipids, fatty acids, organic acids, and phenol amides) were significantly changed. The results of the study provided new ideas regarding the molecular mechanism of the relationship between the yield and quality of *Lycium barbarum* and nitrogen fertilization [17]. Chen et al. used transcriptomics technology to fill the gap of *Lycium barbarum* in SSR markers, assisting in the assessment of *Lycium barbarum*’s genetic diversity, as well as gene mapping and marker-assisted breeding. Single genes were obtained from Illumina sequencing reads and assembled, and then genes in the flavonoids and taurine biosynthesis pathway were found and verified via RT-qPCR. Then, from these single genes, a random selection was marked with EST-SSR markers to verify the quality of the assembly. The PCR amplification of the genomic DNA of the *Lycium barbarum* varieties was successful, and the genetic similarity of the *Lycium barbarum* varieties was shown to be in the range of 0.50–0.99 by polymorphism [18]. Wei et al. combined transcriptomics and metabolomics to reveal that the synthesis and accumulation of carotenoids in *Lycium barbarum* differed in terms of the formation of different colors of *Lycium barbarum*, and that color differences between the red and yellow varieties were obvious during the green fruiting stage, the color change stage, and the red ripening stage. They also noted that the color change stage and red ripening stage were key stages for the differences in the carotenoids and colors of the fruits, and that, during ripening, one *Lycium barbarum* carotenoid biosynthesis-related transcription factor (TF), as well as four factors (CMB1-1, WRKY22-1, WRKY22-3, and RAP2-13-like) in the TFs, were found to be potentially regulated in relation to the phytoene synthase gene (PSY16) and the zeta-carotene desaturase gene (ZDS16) [19].

The above studies of *Lycium barbarum* focus on the effects of the fertilization method, the amount of fertilization, and the type of fertilization on the plants’ yield, quality, and micronutrient functions, but the effects of using different types of selenium fertilizers on *Lycium barbarum* fruits remain poorly understood based on the existing transcriptomics and metabolomics studies. This study was conducted by spraying selenium fertilizer on the foliage during the growth period of *Lycium barbarum* and harvesting when the fruit was ripe. Then, we used transcriptomics and metabolomics technology to explore the effect of exogenous selenium on wolfberries’ differential genes and differential metabolites. In this paper, we sought to understand the effects of different kinds of selenium fertilizer on *Lycium barbarum* metabolites, as well as the differential use of said fertilizers. This study identifies selenium sources that are suitable for spraying on *Lycium barbarum* plants, providing a scientific basis for the development of selenium-rich *Lycium barbarum* in Ningxia.

## 2. Results

### 2.1. Screening for Significantly Different Metabolites in Lycium barbarum

We used the important variables in projection (VIP) >1 as the screening criterion. In the sodium selenite group (Figure 1A), a total of 59 differential metabolites were screened in the sprayed water and sodium selenite treatment groups, of which 32 differential metabolites were upregulated and 27 differential metabolites were downregulated. As shown in nano-selenium group (Figure 1B), a total of 111 differential metabolites were screened in the sprayed water and nano-selenium treatment groups, of which 47 differential metabolites were upregulated and 64 differential metabolites were downregulated. In the organic selenium group (Figure 1C), a total of 129 differential metabolites were screened in the sprayed water and organoselenium treatment groups, of which 71 differential metabolites were upregulated and 58 differential metabolites were downregulated. These screened differential metabolites mainly included amino acids and their derivatives, flavonoids, and alkaloids.

### 2.2. Effect of Exogenous Selenium on Substances Related to Amino Acid Metabolism in Lycium barbarum

In Appendix A, differential metabolites related to amino acid metabolism in *Lycium barbarum* are presented. The term “FC” in log2FC stands for fold change, which represents the ratio of expression levels between two samples or groups. Taking the logarithm of this ratio to the base 2 gives the log2FC. The standard for upregulation and downregulation is defined as 0, where positive values indicate upregulated genes and negative values indicate downregulated genes. The general filtering criterion is commonly set as |log2FC| ≧ 1. The sodium selenite treatment and the sprayed water treatment contained six amino acids and their derivatives. Of these, only l-Citruline and l-Lysine were significantly upregulated, while Phenylacetylglycine, 5-Aminovaleric acid, dl-Norvaline, and Proline were significantly downregulated. Only one amino acid, 3-Methy-l-histidine, was significantly upregulated in the nano-selenium treatment and the sprayed water treatment, and it contained 16 amino acid derivatives, among which 5-Aminovaleric acid, dl-Norvaline, l-Valine, l-Norleucine, Phenylacetylglycine, Proline, and l-Aspartic acid were significantly downregulated while l-Ornithine, trans-4-Hydroxy-l-proline, l-Citruline, and Pyrrole-2-carboxylic acid, l-Methionine, l-Kynurenine, l-Pipecolic acid, and l-Lysine were significantly upregulated. The only metabolites detected in the organic selenium treatment and the sprayed water treatment were amino acid derivatives, seven of which were downregulated, including 5-Aminovaleric acid, l-Valine, dl-Norvaline, l-Norleucine, and l-Aspartic acid. In contrast, trans-4-Hydroxy-l-proline, l-Ornithine, l-Citruline, Phenylacetyl-l-glutamine, Glutathione, l-Kynurenine, l-Pipecolic acid, and 10 other amino acids and their derivatives were upregulated.

### 2.3. Effect of Exogenous Selenium on Flavonoid-Metabolism-Related Substances in Lycium barbarum

As shown in Appendix A, four flavone metabolites were screened for the sodium selenite treatment and the sprayed water treatment: Homoorientin, Narcissoside, 7-Hydroxyflavone, and Kaempferol-3-*O*-rutinoside, which were all significantly downregulated. While six flavonoid metabolites were screened, two of them were significantly downregulated, namely, Protocatechualdehyde and Quercetin 3-*O*-neohesperidoside; moreover, four flavonoid metabolites, i.e., Cyanidin, Naringenin, Chalconaringenin, and Swertiajaponin, were significantly upregulated. Nine flavone metabolites were screened for the nano-selenium treatment and the sprayed water treatment, and only two were significantly upregulated, namely, Lonicerin and Swertiajaponin. Quercetagetin, Quercetin 3-*O*-neohesperidoside, Morin, Eriodictyol, Homoeriodictyol, Delphinidin, and 7-Hydroxyflavone were significantly downregulated, and among the flavonoid metabolites, only Naringenin was significantly upregulated. Protocatechualdehyde, Astragalin, Taxifolin, Narcissosid, Rutin, and Kaempferol-3-*O*-rutinoside were downregulated. Five flavones and seven flavonoids with the organic selenium treatment and the water spray treatment were screened, respectively, among which 2,3-Dihydro-2-phenyl-4*H*-benzopyran-4-one, Naringenin, and Chalconaringenin were significantly upregulated, while eight differential metabolites, including Quercitrin and Eriodictyol, were significantly downregulated.

### 2.4. Effect of Exogenous Selenium on Substances Related to the Alkaloid Metabolism

The results of this investigation are shown in Appendix A. Three alkaloids, namely, Ecgonine, Triangularine, and Nicotinamide, were screened in the sodium selenite (SE1) treatment and water spray (CK) treatment, and only Ecgonine was significantly downregulated, while the other two were significantly upregulated. Fourteen alkaloids were screened in the nano-selenium (SE2) treatment and water spray (CK) treatment, nine of which were significantly downregulated, namely, Calystegine A6, Kynurenic acid, Nb-Feruloyltryptamine, 3-Methylindole, Isoquinoline, Tropine acetate, Synephrine, Tryptophan, Vasicine, beta-Alanine, Strychnine, Nicotinamide, (−)-Cinchonidine, and Cinchonaide were significantly upregulated. Moreover, 22 alkaloids were screened in the organic selenium fertilizer (SE3) treatment and the water spray (CK) treatment, among which Methyl (indol-3-yl)acetate, Ecgonine, Nordihydrocapsaicin, 3-Hydroxy-2-methylpyridine, and 12 other alkaloids were significantly downregulated, while the remaining 10 alkaloids, including beta-Alanine, Calystegine A7, and Trigonelline, were significantly upregulated.

### 2.5. Correlation Analysis of Fruit Metabolites of Lycium barbarum

Correlation analysis can help us to measure the degree of correlation between significantly different metabolites and to further understand the interrelationships between metabolites during changes in biological states. As the number of differential metabolites exceeded 50, the top 50 differential metabolites with the largest VIP values were used for the analysis. Narcissoside was positively correlated with proline (proline) and negatively correlated with naringenin (chalconaringenin) in the sprayed sodium selenite treatment compared to the sprayed water treatment (Appendix A). Naringenin showed a positive correlation with narcissoside and a negative correlation with N6-isopentenyladenosine. Homoorientin was positively correlated with protocatechualdehyde. Valine (dl-norvaline) was positively correlated with protocatechualdehyde, homoorientin, 1D-Myo-inositol1, 4-bisphosphate, and uridine 5′-diphospho-d-glucose.

Rutin was positively correlated with narcissoside and negatively correlated with 3-methylhistidine and N6-isopentenyladenosine in the nano-selenium (SE2) treatment compared to the sprayed water treatment (Appendix A). Pyrrole-2-carboxylic acid was negatively correlated with narcissoside and astragalin. Rutin, l-valine, and behenic acid were negatively correlated with narcissoside, astragalin, rutin, l-valine, and behenic acid, and negatively correlated with N6-isopentenyladenosine and l-ornithine. l-ornithine and cytidine 5-monophosphate (cytidylic acid) were positively correlated. Moreover, 5-Aminovaleric acid and narcissoside, astragalin, kaempferol-3-*O*-rutinoside, kaempferol-3-*O*-rutinoside, l-valine, behenic acid, and dl-norvaline were positively correlated, and narcissoside, astragalin, kaempferol-3-*O*-rutinoside, l-valine, behenic acid, and valine-norvaline were positively correlated with N6-isopentenyladenosine, cytidine 5-monophosphate (Cytidine 5), and N6-isopentenyladenosine, as well as monophosphate (cytidylic acid) and pyrrole-2-carboxylic acid. Quercetin 3-*O*-neohesperidoside was positively correlated with Narcissoside, Rutin, and docosanoic acid. Behenic acid was positively correlated with N6-isopentenyladenosine, l-ornithine, cytidine-5-monophosphate (cytidylic acid), cytosine, and pyrrole-2-carboxylic acid. The correlations were negative for cytidine 5-monophosphate (cytidylic acid), cytosine, and pyrrole-2-carboxylic acid.

Naringenin chalconaringenin was positively correlated with N6-isopentenyladenosine in the organic selenium (SE3) treatment compared to the sprayed water (Appendix A). S-(5-adenosyl)-l-homocysteine (S-(5-Adenosy)-l-homocysteine) was negatively correlated with narcissoside (narcissoside), which was negatively correlated with isopentenyladenosine (N6-isopentenyladenosine), naringenin (chalconaringenin), and cytosine (cytosine). (−)-Cinchonidine was negatively correlated with astragalin, rutin, l-valine, and proline, while it was negatively correlated with l-prnithine. Hydroperoxyoctadeca-10 and 12-dienoic acid were positively correlated.

### 2.6. Analysis of the Transcriptomes of Differentially Expressed Genes in Lycium barbarum

#### 2.6.1. RNA-Seq Data Quality Assessment

As shown in Table 1, the transcriptome libraries treated with sprayed water, sodium selenite, nano-selenium, and organic selenium were sequenced to obtain 542,987,188 clean reads through data filtering, with more than 5 G sequenced per sample; the bases of quality up to the Q30 level accounted for more than 93% of the total bases. This indicated that the RNA-Seq sequencing was of high quality, suggesting that the transcripts were consistent with the subsequent analysis.

#### 2.6.2. Differentially Expressed Genes

As shown in the organic selenium group (Figure 2C), the foliar spraying of different types of selenium fertilizers yielded the highest number of differentially expressed genes in the fresh fruit of *Lycium barbarum* plants treated with organic selenium, amounting to 4676, of which nano-selenium groups (Figure 2B) 2584 and 2092 consisted of up- and downregulated genes, respectively. This was followed by the nano-selenium treatment, with 2495 differentially expressed genes. There were 1264 up- and 1231 downregulated genes in the sodium selenite group (Figure 2A), i.e., the fresh fruit of *Lycium barbarum* treated with sodium selenite. The lowest number of differentially expressed genes was detected in the fresh fruit of the *Lycium barbarum* treated with sodium selenite, amounting to 70, with 66 and 4 up- and downregulated genes, respectively.

#### 2.6.3. GO Enrichment Analysis of Differentially Expressed Genes

GO refers to gene ontology, which is a classification system that is widely used in the field of bioinformatics, as well as in the biological field in general. Cellular components, molecular function, and biological process are the three branches of GO. GO enrichment analysis was performed on the selected differential multiplicity |log2(FC)|>7 genes, and the top 30 GO terms that were significantly enriched were analyzed. As shown in sodium selenite group (Figure 3A), the differential genes between the water spray and sodium selenite treatments were mainly involved in molecular function and biological processes, with molecular function accounting for the majority of them. In total (GO:0004601), 41 differential genes were involved in catalytic activity (GO:0003824), and 22 differential genes were found to be involved in oxidoreductase activity (GO:0016491). In the nano-selenium group (Figure 3B), the differential genes between the water spray and the selenium nano-treatment were mainly involved in molecular functions and biological processes, of which 273 differential genes were involved in the organonitrogen compound metabolic process (GO:1901564), 121 differential genes were involved in the phosphorus metabolic process (GO:0006793), and 322 differential genes were involved in protein binding (GO:0006793). Among them, 273 differential genes were involved in the organonitrogen compound metabolic process (GO:1901564), 121 differential genes were involved in the phosphorus metabolic process (GO:0006793), 322 differential genes were involved in protein binding (GO:0005515), 197 differential genes were involved in adenyl ribonucleotide binding (GO:0005515), and 197 differential genes were involved in molecular functions and biological processes. Adenyl ribonucleotide binding was also relevant (GO:0032559). In the organic selenium group (Figure 3C), the differential genes between the water spray and the organoselenium spray treatments were mainly involved in molecular functions and biological processes. Among them, 557 differential genes were involved in the organonitrogen compound metabolic process (GO:1901564), 662 differential genes were involved in protein binding (GO:0005515), and 738 differential genes were involved in catalytic activity (GO:0003824).

#### 2.6.4. Results of KEGG Enrichment Analysis of Differentially Expressed Genes

To understand the functions and major signaling pathways of the differential genes, KEGG pathway analysis was performed on the selected |log2FoldChange|>7 differential genes. As shown in the sodium selenite group (Figure 4A), the differential genes between the water spray and sodium selenite spray treatments were significantly enriched in 13 pathways, which were mainly involved in phenylpropanoid biosynthesis, metabolic pathways, and the biosynthesis of secondary metabolites. In the nano-selenium group (Figure 4B), the differential genes between the water spray and selenium nanospray treatments were significantly enriched in 20 pathways, including plant–pathogen interactions, carbon metabolism, and the biosynthesis of amino acids. In the organic selenium group (Figure 4C), the differential genes between the water and selenium spray conditions were significantly enriched in 20 pathways, mainly those related to carbon metabolism, the biosynthesis of amino acids, and homologous recombination.

#### 2.6.5. Metabolomics and Transcriptomics Association Analysis

KEGG enrichment associations were used to calculate the significance level of gene and metabolite enrichment in each pathway by performing a comparative analysis of pathway enrichment in the transcriptome and metabolome. The differential genes obtained through the comparison of the water spray and sodium selenite and the differential metabolites shared five common metabolic pathways (sodium selenite group, Figure 5A), including the biosynthesis of secondary metabolites, metabolic pathways, glutathione metabolism, flavonoid biosynthesis, and amino sugar and nucleotide sugar metabolism. The differential genes, as well as the differential metabolites obtained in the comparison of the water spray and the selenium nanoparticles, had 10 common metabolic pathways (nano-selenium group, Figure 5B). These included the arginine and proline metabolisms, the histidine metabolism, pentose and glucuronate interconversions, the tyrosine metabolism, aminoacyl-tRNA biosynthesis, the cysteine and methionine metabolisms, folate biosynthesis, the carbon metabolism, the starch and sucrose metabolisms, and the tryptophan metabolism pathways. The differential genes, as well as the differential metabolites, which were obtained through the comparison of the water spray and organoselenium, had 10 common metabolic pathways (organic selenium group, Figure 5C), including pathways such as the starch and sucrose metabolism, tryptophan metabolism, purine metabolism, arginine and proline metabolism, linoleic acid metabolism, pentose and glucuronate interconversions, carbon fixation in photosynthetic organisms, pentose and glucuronate interconversions, lysine degradation, and ascorbate and aldarate metabolism.

## 3. Discussion

Selenium-enriched *Lycium barbarum* uses the plant’s own metabolic functions of exogenous selenium absorption and conversion before being enriched in the fruit, so that the selenium content of *Lycium barbarum* berries is significantly increased. Different types of exogenous selenium are related to different selenium-enrichment effects. A variety of bioactive substances, including amino acids, flavonoids, alkaloids, etc., confer many functions, such as immune regulation, antioxidant properties, and beauty and skincare benefits, to *Lycium barbarum* [20,21,22,23].

Some studies have found that, by spraying different types and concentrations of selenium fertilizers, the content of plant metabolite amino acids can be increased to some extent. For example, Huang et al. found that the application of selenium fertilizers led to an increase in the level of metabolite amino acids compared to the spraying of clear water and nano-selenium [24]. Li et al. found that selenium fertilizers were effective in terms of increasing amino acid content, with selenite having a more pronounced ability to increase amino acids than selenate [25]. The metabolic results achieved by spraying selenium fertilizer treatments on the foliage of plants in a study by Cheng et al. showed that Se promotes the metabolism of amino acids, which increases the content of amino acids in plants [26]. Similarly, in our study, the use of different exogenous sources of selenium (100 mg·L^−1^, sodium selenite, nano-selenium, and organic selenium) produced differential metabolites of amino acids and their derivatives in fruits obtained by foliar spraying; there were 17 differential metabolites in the organic selenium treatment group, with 7 showing downregulation and 10 upregulation, and 16 in the nano-selenium treatment group, of which 7 were downregulated and 9 were upregulated. Only six were increased in the sodium selenite treatment group and only two were upregulated, while the remaining four were significantly downregulated. By spraying plants with three selenium fertilizers, the contents of amino acids and their derivatives increased to varying degrees. This finding is in line with the results found in studies of the scholars mentioned above. We also determined that the increase in the content of amino acids and their derivatives was most pronounced in the organic selenium and nano-selenium treatment groups.

Flavonoids belong to a group of low-molecular-weight phenolic compounds, the content of which directly affects superoxide radical scavenging, among other effects, which can regulate bodily functions and enhance the body’s resistance to disease [27,28,29]. Many scholars have found that, by spraying different types and concentrations of selenium fertilizers, the content of flavonoids in the plant body is increased to some extent. Deng et al. found that, after spraying selenite at concentrations of 0.4 mM, 0.8 mM, 1.6 mM, and 3.2 mM onto leaves, the total flavonoids were increased to varying degrees. The most significant increases were found at 0.8 mM and 1.6 mM, where the flavonoid contents increased by 19.26% and 43.99%, respectively [30]. Ahmad et al. sprayed 10 ml (Se 50µM) of selenium nutrient solution onto plant leaves and found that an accumulation of flavonoid content occurred [31]. Li et al. improved the antioxidant defense system of tea trees by spraying 0, 2.5, 5, 10, and 20 mg/L of nano-selenium onto a tea tree, which increased the accumulation of flavonoids while promoting the levels of secondary metabolites [32]. Hu et al. found that the flavonoid biosynthesis genes of tea trees were significantly enriched in the group treated with 0.5 mg/L of selenium, demonstrating that the application of selenium can promote flavonoid synthesis [33]. Similarly, in our study, flavonoid metabolites within the fruit were screened differently after spraying different selenium fertilizers on *Lycium barbarum*, and a comparison with the sprayed water revealed that the most flavonoid metabolites were screened after spraying organic selenium, followed by nano-selenium, with sodium selenite producing the fewest. Based on the different selenium sources which were sprayed, the flavonoid contents within the fruits also increased to some extent, a finding that concurs with those of the studies mentioned above.

Alkaloids are important secondary metabolites that prevent some diseases by free radical scavenging or by binding to the catalysts of oxidative reactions [34]. Numerous studies have found that, by spraying selenium fertilizers on crop leaves, the alkaloid content in the crop can be increased to some extent [35,36,37]. In this study, it was found that, after spraying sodium selenite, nano-selenium, and organic selenium, alkaloid-related metabolites were screened in wolfberry fruits with 3, 9, and 22 metabolites, respectively; this is in line with the results of the above studies, which showed that alkaloids were increased to different degrees in fruits after spraying these selenium fertilizers. Meanwhile, different selenium fertilizers screened different related metabolites, and this result may help the exploitation of *Lycium barbarum* in other areas.

Complex regulatory processes occur within organisms [38]. In our study, the differentially expressed genes (DEGs) and differentially expressed metabolites (DEMs) in the three treatment groups (sodium selenite—SE1, nano-selenium—SE2, and organic selenium—SE3 versus sprayed water control—CK) were involved to varying degrees in glutathione metabolism. Among the treatment groups, SE3vsCK had the highest number of DEGs (19) and DEMs (2) involved in glutathione metabolism, followed by SE2vsCK, with 7 DEGs and 1 DEM. SE1vsCK had the fewest, with 2 DEGs and 1 DEM. Research has shown that glutathione metabolism is regulated by genes such as GSTM1 and GSTP1, which produce glutathione cyclic enzymes and play an important role in multiple pathways, including plant signal transduction and defense against biotic stress [39,40,41,42].

The biosynthesis of secondary metabolites roughly involves amino acids, alkaloids, flavonoids, and phenols, which act as defense molecules in plants [43]. Tryptophan, tyrosine, lysine, arginine, and ornithine are the main sources of alkaloids, including terpenoid indole alkaloids, tropane alkaloids, and purine alkaloids, all of which protect plants from environmental threats [44]. Flavonoids are the main components synthesized through enzymatic reactions with, for example, chalcone synthase (CHS), which catalyzes the production of coumaroyl-CoA and upregulates metabolites such as naringenin chalcone [45,46,47]. This indicates that coumaroyl-CoA stimulates the synthesis of flavonoids, enhancing their ability to scavenge free radicals, exhibit antioxidant properties, and produce anti-inflammatory effects. This is consistent with our research findings: the application of selenium fertilizers enhances the activity of coumaroyl-CoA. In the SE2vsCK and SE3vsCK treatment groups, the differential expression of 4-coumarate–CoA ligase was upregulated, leading to the accumulation of metabolites such as naringenin and chalconaringenin and an increase in flavonoid synthesis. In the SE2 and SE3 treatment groups, l-pipecolic acid and l-ornithine were upregulated, promoting the accumulation of amino acids and their derivatives, which was not observed in the SE1 treatment group. Similarly, the upregulation of scopadulcic acid and beta-Alanine in the SE2 and SE3 treatment groups promoted the accumulation of alkaloids, which was not found in the SE1 treatment group. Therefore, it can be inferred that, after the application of different types of selenium fertilizers, the SE1, SE2, and SE3 treatment groups showed varying degrees of upregulation in different biosynthetic and metabolic pathways. The upregulation in SE2 and SE3 was higher than in SE1, suggesting that SE2 and SE3 are effective approaches for improving the quality of *Lycium barbarum*.

## 4. Materials and Methods

### 4.1. Study Region

The trial was conducted at the Ningxia Garden Test Site (38°40′4″ N,106°9′31″ E), located in the northwestern suburbs of Yinchuan City. The area is about 1110 m above sea level and has a typical temperate monsoon/arid continental climate. The climate is dry and rainy at various points in the year, with an average annual precipitation of about 200 mm. The annual average temperature is only 8.8 °C, while the average temperature difference between day and night is about 10–15 °C. The annual total accumulated temperature exceeds 210 °C and can reach 3000 °C or above, the annual sunshine hours exceed 2800 h, the frost-free period is about 150–170 days, and the water evaporation is strong. The soil in this area is light lime silica with a heavy texture.

### 4.2. Study Methods

A total of 216 7-year-old *L. barbarum* plants were utilized for this research, as demonstrated in Figure 6. The plants were allocated within 12 experimental areas, consisting of four distinct treatments: sprayed water (CK), sodium selenite (SE1), nano-selenium (SE2), and organic selenium (SE3). Each treatment comprised two rows containing a total of 18 plants. The experiment was conducted using a completely randomized block design, which was replicated three times.

This experiment required the appropriate application of selenium fertilizer using the conventional drip irrigation fertilization method (urea 0.45 t, ammonium dihydrogen phosphate 0.45 t, potassium dihydrogen phosphate 1 t). The selenium sources used for testing were sodium selenite (SE1), nano-selenium (SE2), organic selenium (SE3), and sprayed water (CK). The selenium fertilizer concentration reference was set to 100 mg·L^−1^ [48]. The fertilizer was sprayed every 15 days, starting from June 1 with the appearance of the first fruit, and the treatment ended on June 30 with the first fruit picking, for a total of three sprays. The selenium content of the fertilizers is shown in Table 2.

### 4.3. Sample Collection

Five Lycium trees with a uniform plant type, growth potential, and fruit load were randomly selected in each experimental plot, and large, medium, and small as well as sun- and shade-oriented fruits were randomly picked from each selected tree to form an average sample. These fruits were picked up using clean forceps by a researcher wearing gloves, and were quickly placed into marked lyophilization tubes. Then, the lyophilization tubes were immediately put into boxes containing liquid nitrogen and transported back to the laboratory for storage in a –80 °C refrigerator for the metabolome and transcriptome experiments.

### 4.4. Experimental Methods

#### 4.4.1. Metabolome Analyses

The metabolomics analysis comprised fifteen samples divided into five groups, namely, CK, QC, SE1, SE2, and SE3; each group had three biological replicates, of which QC was a quality control sample. For metabolite extraction, *Lycium barbarum* fruits were vacuum-dried and frozen in a lyophilizer (Scientz-E100 F, Ningbo, China) for 24 h. The fruits were ground (60 Hz, 30 s) to a powder using a grinder (MM 400, Retsch, Haan, Germany). The berry powder was extracted with 0.7 ml of aqueous methanol solution (3:1) at 4 °C overnight. The powder was centrifuged at 12000 rpm (centrifugal force, 13,800× *g*, radius, 8.6 cm) for 15 min, and the supernatant was aspirated and filtered through a microporous membrane (pore size = 0.22 μm) for the UPLC-MS/MS analysis. The analytical conditions were as follows: the UHPLC separation was carried out using an EXIONLC System (Sciex, Darmstadt, Germany). The mobile phase A comprised 0.1% formic acid in water, and the mobile phase B was acetonitrile. Sample measurements were performed with a gradient program that employed starting conditions of 98% of solvent A and 2% of solvent B. Within 10 min, a linear gradient to 50%A/50% B was programmed, and 5%A/95%B was maintained for 2 min. Subsequently, a composition of 98%A/2%B was adjusted at 0.1 min and maintained for 2.9 min. The column oven was set to 40 °C, and the injection volume was 2 μL, while the flow rate of the column was 400 μL/min. A Sciex QTrap 6500+ (Sciex Technologies, Framingham, MA, USA) was used for the assay development. Typical ion source parameters were as follows: ion spray voltage: +5500/-4500 V, curtain gas: 35 psi, temperature: 400 °C, ion source gas 1: 60 psi, ion source gas 2: 60 psi, and DP: ± 100 V. A specific set of MRM transitions was monitored for each cycle based on the metabolites eluted during that cycle. The metabolite identification criteria were confirmed based on RT vs. Q1/Q3. The RT was within one second and Q1/Q3 had an error of 0.2-0.4.

For differential metabolite screening, multivariate statistical analysis of the metabolites was carried out using orthogonal partial least squares discriminant analysis (OPLS-DA). By combining the variable importance in project (VIP) of the OPLS-DA model, metabolites that differed between treatments could be initially screened. Then, the different metabolites were further screened by combining the difference multiplier values; finally, the metabolites with a fold change of ≥2, or a fold change of ≤0.5 and a VIP of ≥1, were selected as the differential metabolites.

#### 4.4.2. Transcriptomics Analysis

The transcriptomics analysis comprised twelve samples divided into four groups, namely, CK, SE1, SE2, and SE3, with three biological replicates in each group. For RNA extraction and sequencing, after the samples were ground into powder, total RNA was extracted from the fruits using TRIzol (Invitrogen, Carlsbad, CA USA), followed by a treatment with RNase-free DNase I (Takara, Tokyo, Japan). The RNA quality assessment was performed using a Nanodrop ND-1000 spectrophotometer (Thermo Scientific, Waltham, MA, USA) for RNA purity determination. Each sample required 1.5 μg of RNA for library construction, according to the instructions for the NEBNext^®^ Ultra™ RNA Library Prep Kit for Illumina^®^ (NEB, Dallas, TX, USA). In brief, the following steps were performed: first, eukaryotic mRNA was enriched with magnetic beads with Oligo (dT) by binding to the polyA tail of mRNA through A-T complementary pairing. The mRNA was subsequently broken into short fragments by adding fragmentation buffer, and first-strand cDNA was synthesized using a six-base random primer in the presence of M-MuLV reverse transcriptase, followed by the second-strand cDNA using DNA polymerase I and dNTPs. Double-strand cDNA was purified using an AMPure XP system (Beckman Coulter, Beverly, MA, USA). cDNA fragments of 150–200 bp were preferentially selected using the AMPure XP system (Beckman Coulter, Beverly, MA, USA), and, finally, the library quality was assessed using an Agilent Bioanalyzer 2100 system.

The Illumina sequencing principle schematic is shown in Figure 7. During this process, after the library is qualified, it is pooled according to its effective concentration and the desired data volume, and finally, the corresponding model of the sequencer is selected for online sequencing according to the sequencing read length. The basic principle of this type of sequencing is sequencing by synthesis. Four types of fluorescently labeled dNTP, DNA polymerase, and junction primers are added to the sequencing flow cell for amplification. When extending the complementary strand of each sequencing cluster, each fluorescently labeled dNTP that is added releases a corresponding fluorescence. The sequencer captures the fluorescence signals and converts the light signals into sequencing peaks through computer software, thus obtaining the sequence information of the fragment to be tested.

### 4.5. Data Processing

#### 4.5.1. Metabolomic Data Analysis

The SCIEX Analyst Work Station Software (Version 1.6.3) was employed for MRM data acquisition and processing. For the quantitative analysis of the samples, a valuation method was devised by integrating the peak area of the sample to produce a quantitative value. Hierarchical cluster analysis (HCA), principal component analysis (PCA), and partial least squares discriminant analysis (OPLS-DA) were applied to the metabolites in R language (www.r-project.org/ (accessed on 15 September 2021)) to investigate the accumulation characteristics of the metabolites under the different selenium treatments. The KEGG website (http://www.kegg.jp/ (accessed on 15 September 2021)) was used to perform the differential metabolite passage enrichment analysis.

#### 4.5.2. Transcriptome Data Analysis

For data quality management, the raw data in the fastq format were first processed using a Perl script to remove splice reads, N (unknown base) reads, and low-quality reads, and to calculate the Q20, Q30, and GC contents of the clean reads. Differential gene analysis was performed using DESeq software (1.10.1) [49], based on a negative binomial distribution model, to calculate differential genes between samples. The BH method was used to adjust the resulting *p*-values to control the false discovery rate.

#### 4.5.3. Differential Gene GO Enrichment and KEGG Pathway Analysis

Gene ontology (GO, http://www.geneontology.org/ (accessed on 1 September 2021)) is an international standard classification system for gene function. Using GOSe (Release 2.12) software, the hypergeometric distribution of differential genes in relation to a particular branch of the GO classification was calculated, and a specific *p*-value was obtained via hypothesis validation, with *p* < 0.05 indicating that the differential genes were enriched in that GO. KEGG (the Kyoto Encyclopedia of Genes and Genomes) (http://www.genome.jp/kegg/ (accessed on 1 September 2021)) is a database for the systematic analysis of gene function and genomic information. KOBAS v2.0 software was used to calculate the enrichment of differential genes in the KEGG pathway.

#### 4.5.4. Combined Analysis of Metabolomic Data and Transcriptomic Data

Differential genes and differential metabolites were analyzed using the Spearman rank and rank correlation methods, and the results were constructed as correlation heat maps with a correlation coefficient *p*-value < 0.05 as the condition for a significant correlation. The pathway analysis was based on the KEGG website and programmed in the Perl language for joint analysis.

## 5. Conclusions

The foliar application of selenium fertilizer on *Lycium barbarum* leaves achieved the effective regulation of metabolites such as amino acids, flavonoids, and alkaloids in *Lycium barbarum* by influencing the expression of differentially expressed genes and metabolites. In particular, organic selenium and nano-selenium showed more pronounced regulatory effects on differentially expressed genes and metabolites in *Lycium barbarum*. These research findings provide important reference value for this biological study on promoting the growth of *Lycium barbarum* using selenium fertilizer. Our findings also broaden the research direction for further exploration of the application of selenium fertilizer in other plant species. The rational formulation of treatment measures in this experiment provides a design strategy to effectively improve the quality of *Lycium barbarum* and other crops.

## Figures and Tables

**Figure 1 molecules-28-08088-f001:**
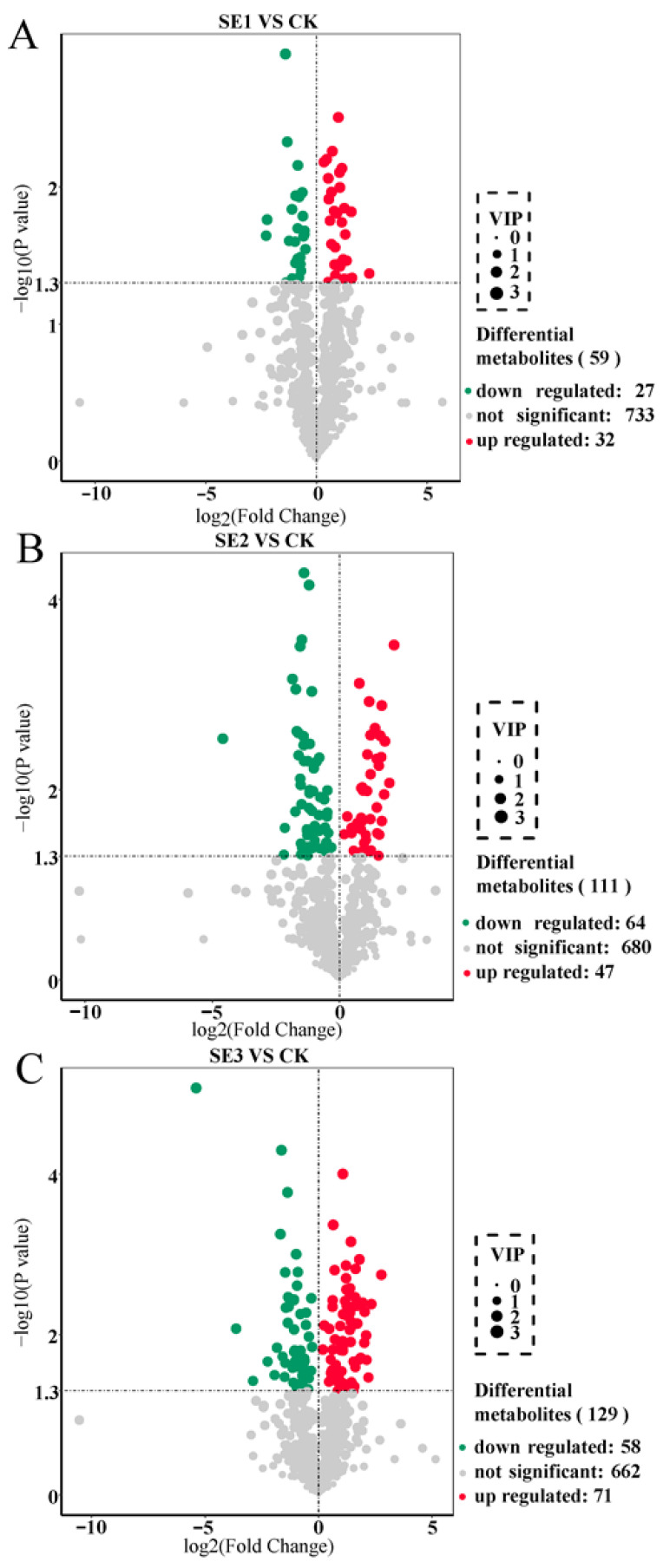
Volcano plot of differential metabolites in *Lycium barbarum* fruit samples ((**A**): sodium selenite (SE1), (**B**): nano-selenium (SE2), and (**C**): organic selenium (SE3), versus sprayed water control (CK)).

**Figure 2 molecules-28-08088-f002:**
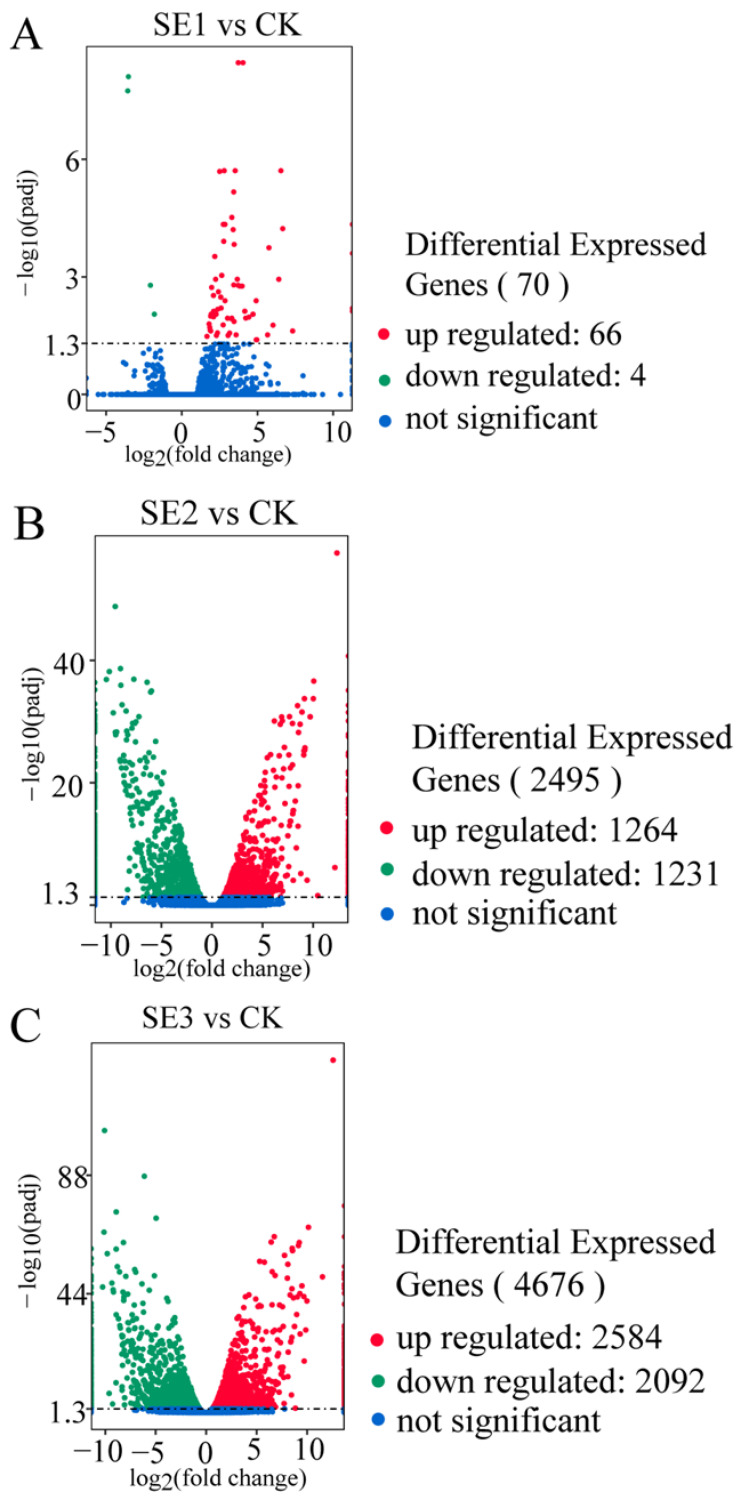
Volcano map of differentially expressed genes in *Lycium barbarum* samples ((**A**): sodium selenite (SE1), (**B**): nano-selenium (SE2), and (**C**): organic selenium (SE3), versus sprayed water control (CK)).

**Figure 3 molecules-28-08088-f003:**
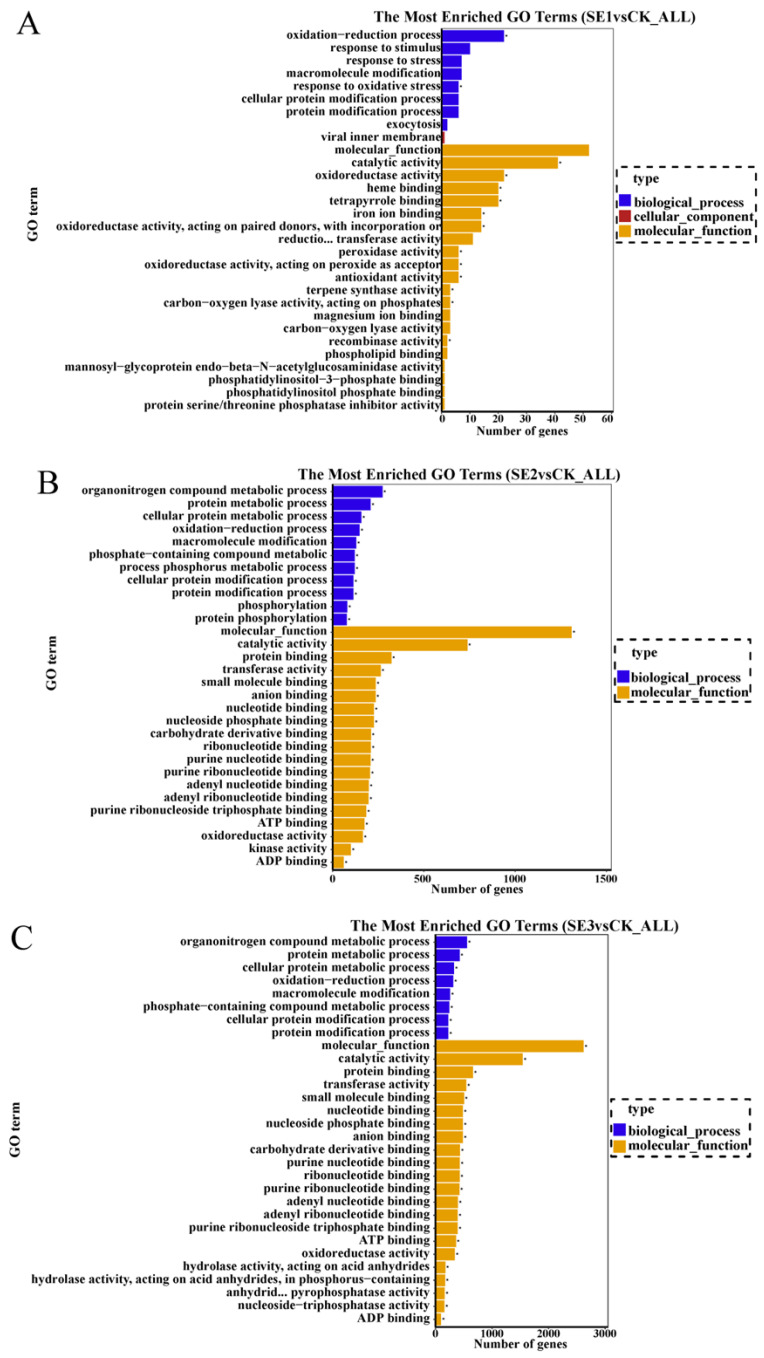
GO enrichment bar chart of differentially expressed genes in *Lycium barbarum* samples: ((**A**): sodium selenite (SE1), (**B**): nano-selenium (SE2), and (**C**): organic selenium (SE3), versus sprayed water control (CK)).

**Figure 4 molecules-28-08088-f004:**
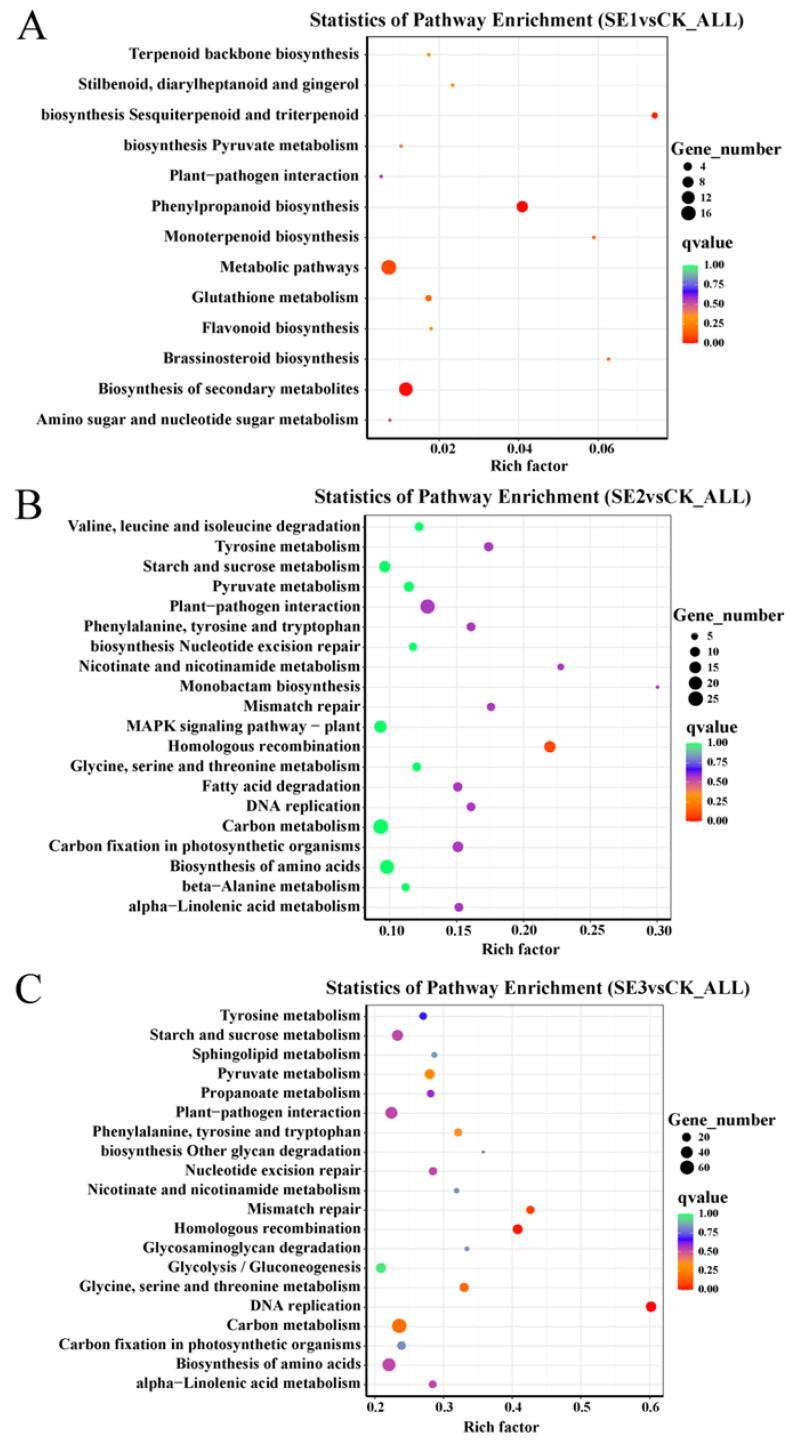
Analysis of KEGG pathways for differential genes in *Lycium barbarum* samples: (**A**): sodium selenite (SE1), (**B**): nano-selenium (SE2), and (**C**): organic selenium (SE3), versus sprayed water control (CK).

**Figure 5 molecules-28-08088-f005:**
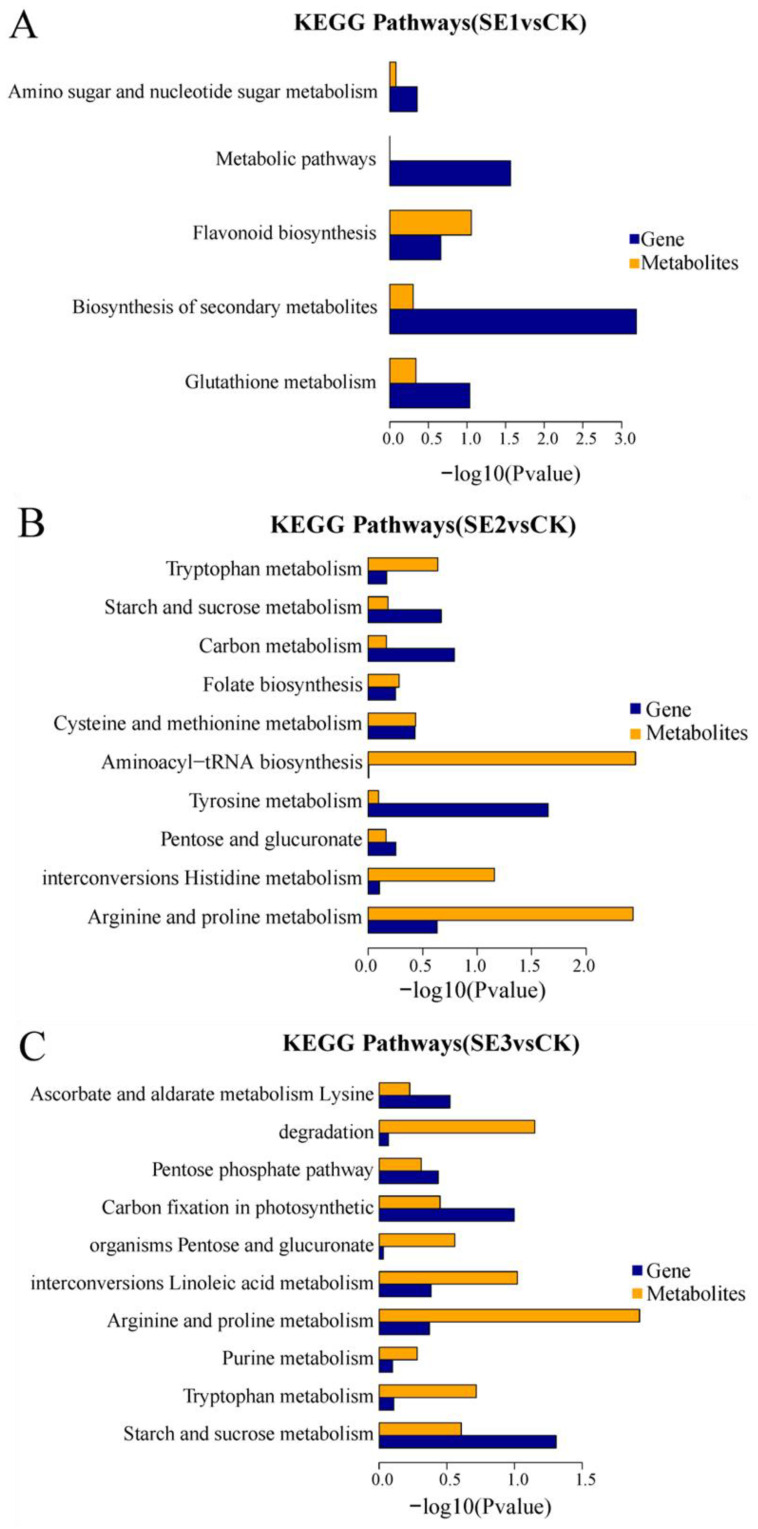
KEGG enrichment histogram of differential metabolites and differential genes in *Lycium barbarum* samples. (**A**): sodium selenite (SE1), (**B**): nano-selenium (SE2), and (**C**): organic selenium (SE3), versus sprayed water control (CK).

**Figure 6 molecules-28-08088-f006:**
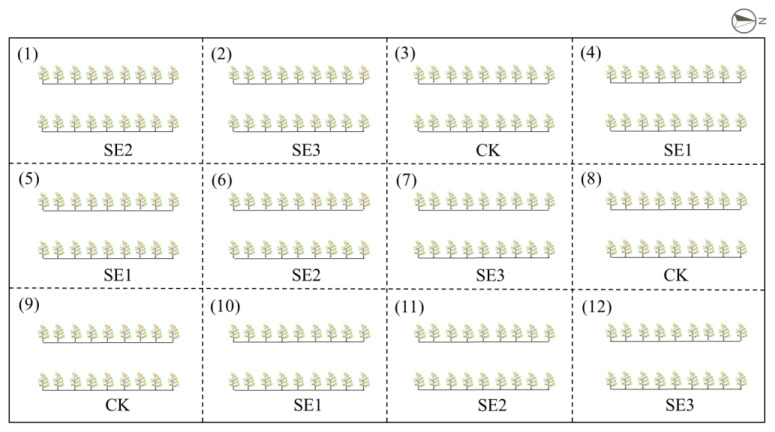
Layout of the field trial.

**Figure 7 molecules-28-08088-f007:**
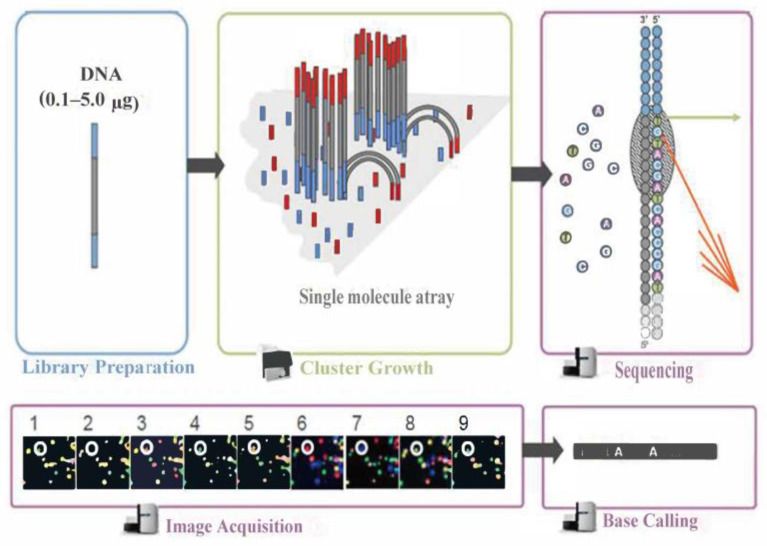
Illumina sequencing principle schematic.

**Table 1 molecules-28-08088-t001:** Sequencing data statistics.

Sample Name	Raw Reads	Raw Bases	Clean Reads	Clean Bases	Error Rate	Q20	Q30	GC Content
CK-1	45,330,910	6.79 G	44,340,314	6.65 G	0.03%	97.82%	93.72%	42.53%
CK-2	48,021,620	7.2 G	47,150,434	7.07 G	0.03%	97.82%	93.71%	42.44%
CK-3	41,223,694	6.18 G	40,393,914	6.06 G	0.03%	97.94%	93.99%	42.51%
SE1-1	46,940,776	7.04 G	46,199,336	6.93 G	0.03%	97.91%	93.89%	42.56%
SE1-2	46,966,566	7.04 G	46,364,144	6.95 G	0.03%	97.91%	93.85%	42.40%
SE1-3	40,378,554	6.05 G	39,796,962	5.97 G	0.03%	97.87%	93.82%	42.45%
SE2-1	44,866,144	6.72 G	44,070,522	6.61 G	0.03%	97.62%	93.28%	42.20%
SE2-2	53,107,194	7.96 G	52,151,418	7.82 G	0.03%	97.82%	93.74%	42.28%
SE2-3	49,901,306	7.48 G	49,416,786	7.41 G	0.03%	97.83%	93.72%	42.60%
SE3-1	41,495,698	6.22 G	40,914,436	6.14 G	0.03%	97.90%	93.87%	42.48%
SE3-2	45,808,966	6.87 G	44,980,762	6.75 G	0.03%	97.86%	93.81%	42.44%
SE3-3	47,927,700	7.18 G	47,208,160	7.08 G	0.03%	97.96%	94.02%	42.39%

Notes: Raw reads, raw bases: the number of reads and the total number of bases (in G) of the original sequencing data. Clean reads, clean bases: the number of reads and the number of bases (in G) obtained after the completion of QC. Error rate: the average sequencing error rate of clean reads. Q20, Q30: the ratio of the number of bases with Phred quality values greater than 20 and 30 to the total number of bases (clean data). GC content: the ratio of GC to the total number of bases (clean data) in the sequencing data.

**Table 2 molecules-28-08088-t002:** The test fertilizers.

Fertilizer Types	Selenium Content	Fertilization Concentration	Solubility	Water Solubility
Sodium selenite, SE1	44.70%	100mg/L	95.5%	Soluble in water
Nano selenium, SE2	4.00%	100mg/L	99.0%	Soluble in water
Organic selenium, SE3	0.60%	100mg/L	99.0%	Extremely soluble in water

## Data Availability

Data are contained within the article and Appendix A.

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
