# Peer review of "Combined Transcriptomics and Metabolomics Analysis Reveals the Effect of Selenium Fertilization on Lycium barbarum Fruit"

_molecules, 2023, doi:10.3390/molecules28248088_

Round 1

Reviewer 1 Report

Comments and Suggestions for Authors

-This paper is multi-omices study for combined transcriptomics and metabolomics analysis reveals the effect of selenium fertilization on Lycium barbarum fruit, which is an interesting study.

- In this study, authors performed quantitative analysis of the metabolites? It should be presented as a Table.

-In addition, the univariate statistical analysis results suggest that the understanding of the metabolomics results will be higher.

-In this study, it is necessary to supplement the explanation of the merits compared to the case where only a single omics was performed on the results obtained through the integrated omics study.

Comments on the Quality of English Language

English correction is needed.

Reviewer 2 Report

Comments and Suggestions for Authors

Define LBP abbreviation in Abstract and after its first appearance in text, please.

"et al." must be typed in italics, please correct in whole manuscript.

Line 103-105 -> Sentence is not clear. Please review to avoid redundancy of ideas.

Lines 110 - 119 -> Specify properly in text what is shown in Figure 1 -> sodium selenite group (Figure 1A), nano-selenium group (Figure 1B) and organoselenium group (Figure 1C).

Figure 1 title -> Lycium barbarum -> must be typed in italics, correct please - Same for Figures 2 to 6. Please, verify and correct whole manuscript! 

Figure 1 legend -> Please review legend to become more assertive and clear. Name tested groups in legend please (1A for sodium selenite group, 1B for nano-selenium group and 1C for organoselenium group). Also, include method used to identify the metabolites. Same for Figures 2 to 6. 

Table 1 title -> use Lycium barbarum instead of wolfberries or altogether; Same for Tables 2 and 3.

Table 1 -> Specify treatment groups -> SE and CK acronyms must be written in main text; What is named in first column of the table, samples or treatments? Table naming for type of selenium used must be clearer -> Same for Tables 2 and 3 and respective text.

Line 131 -> Why amino acid derivatives names are repeated here? Improve text conciseness, please.

Line 127-146 -> What was the criterion or cut-off for be considered as up-regulated or down-regulated here? Mention it please in text; Why 3-methyl-L-histidine with log2(FC)=1.5540 is considered significantly up-regulated and L-lysine with log2(FC)=1.6646 is not? Include statistics in Table 1, please. Review this whole paragraph to describe results correctly. -> Same for Tables 2 and 3 and respective text.

Line 134 -> "It contained 15 amino acid derivatives..." -> Which sample treatment is this? Name it properly in text. Also, there are no treatment sample with 15 amino acid derivatives. SE3 vs CK has 17 amino acid derivatives listed in Table 1, right? Please, review and correct.

Figure 4 -> Use the same color pattern for GO type in all figures A, B and C.

Figure 6 -> Depict/indicate in graphs A, B and C what is each group/selenium type please.

Line 346 -> What is ENMS?

Line 351 -> "Only five were increased..." -> ??? Do you mean differentially presented?

Discussion -> Transcriptomics results and its association with metabolomics is lacking. For instance, why such discrepancy - more than 20x - in GO analysis regarding number of expressed genes within the three treatments? Please include some commentary or explanation for that.

Line 407 and 416 -> Extra space after (SE1), correct.

Line 434 -> Add space after Hz.

Line 441 -> Acetonitrile -> acetonitrile, correct.

Line 448 -> oC -> oC, correct.

Line 465 -> Add space after NEB,

Line 469 -> What is M???

Line 475 -> DNA sequencing? How was it done?

Conclusion -> Should be reviewed because as written is more like a summary of the results. What can you conclude with the obtained results?

Round 2

Reviewer 2 Report

Comments and Suggestions for Authors

1. The following paragraph at introduction remains unclear: "Moreover, we sought to understand the effects of metabolite substances of different kinds of selenium fertilizer on the wolfberry, as well as the differential use of the substance." -> What substance? Metabolite or selenium fertilizer? Please review.

2. All figure legends must be reviewed again. It is redundant, repetitive and unclear as stated: "(1A for sodium selenite group, 1B for nano-selenium group and 1C for organoselenium group), (SE1-Sodium selenite, SE2-Nano-selenium group , SE3-Organic selenium, CK-Sprayed water.)" Try something like: " 1A-Sodium selenite (SE1), 2A-Nano-selenium (SE2), 3A-Organic selenium, versus sprayed water control (CK)".

3. Lycium barbarum typing remains to be corrected to italics at line 79.

4. Citation of references throughout text must be reviewed and corrected according to journal guidelines. It appears differently as: White et al. (line 59), Shi Z et al. (line 72), Chen, C. et al. (line 79) or (Anders et al., 2010) (line 545), so standardize it please.

5. Correct typing of the word "classification" in Tables 1 and 2.

6. "As shown in Table 2, Differential metabolites" (line 136), is this correct? This should be Table 1, right? Review uppercase typing usage, please. For instance, at Table 2 sometimes is "Flavonoids" other "flavonoids", standardize it please.

7. SE2 data description in text (lines 143-151) is still discordant of what is presented in Table 1, please review. Is not L-methionine or proline also an amino acid? It should be better keep as presented before, just correcting to 16 the number of metabolites found. Also, why L-methionine, L-kynurenine, L-pipecolic acid and L-lysine are not mentioned in text as upregulated? Review it please.

8. Some extra spacing and commas were found in text (e.g., at lines 114, 172, 309), please review whole manuscript and correct. Also, several paragraphs are not aligned to the right margin (justified), please review all.

9. "Anders et al., 2010" (line 545) is not listed in references. Check whole manuscript for that, please.

10. Sequencing method description is lacking. Please add how was it done.
